J Physiol 603.6 (2025) pp 1645–1662

# α4β2* nicotinic acetylcholine receptors drive human temporal glutamate/GABA balance toward inhibition

Katiuscia Martinello[1] , Addolorata Mascia[2], Sara Casciato[2], Giancarlo Di Gennaro[2],
Vincenzo Esposito[2,3], Michele Zoli[4], Cecilia Gotti[5] and Sergio Fucile[2,6]

[1]*Department of Human, Social & Health Sciences, University of Cassino and Southern Lazio, Cassino, Italy*
[2]*IRCCS Neuromed, Pozzilli, Italy*
[3]*Department of Human Neurosciences, Sapienza University of Rome, Rome, Italy*
[4]*Department of Biomedical, Metabolic and Neural Sciences, Center for Neuroscience and Neurotechnology, University of Modena and Reggio Emilia, Modena, Italy*
[5]*CNR Institute of Neuroscience, Vedano al Lambro, Italy*
[6]*Department of Physiology and Pharmacology, Sapienza University of Rome, Rome, Italy*

The peer review history is available in the Supporting Information section of this article (https://doi.org/10.1113/JP285689#support-information-section).

**The Journal of Physiology**

**Abstract figure legend** α4β2* nicotinic acetylcholine receptors (α4β2*nAChR) modulate the synaptic transmission in the temporal human neocortex, recorded from layer 5 pyramidal neurons. We describe a monosynaptic mechanism at the GABAergic synapses (left), in which the activation of the heteromeric nicotinic receptors, expressed on the presynaptic cell, enhances the firing rate of interneurons, thus increasing GABA release. We also report a disynaptic mechanism occurring at the glutamatergic synapse (right). Here, α4β2*nAChR induces the activation of presynaptic GABA_B receptors on glutamatergic terminals, reducing glutamate release. Both mechanisms increase the general inhibitory tone in the cortex.

**Katiuscia Martinello** is an Associate Professor of Physiology at the Department of Human, Social, and Health Sciences, University of Study of Cassino and Southern Lazio (Italy). Following her PhD, she has been working on the modulation of synaptic neurotransmission in humans and rodents, highlighting the difference between healthy and pathological conditions. In the present study, she has shown how modulating neuronal heteromeric nicotinic receptors can increase the inhibitory tone in the human neocortex.

The Journal of Physiology

**Abstract** Heteromeric nicotinic acetylcholine nAChRs (nAChRs) containing the $\alpha 4$ and $\beta 2$ subunits ($\alpha 4\beta 2^*$ nAChRs) modulate neurotransmitter release in several regions of the brain. In temporal lobe epilepsy, inhibitory GABAergic neurotransmission is altered, whereas no evidence of nicotinic dysfunction has been reported. Here, we investigated, in human epileptic cortical tissues, the ability of $\alpha 4\beta 2^*$ nAChRs to modulate synaptic transmission. An increased expression of $\alpha 4$ and $\beta 2$ subunits was observed in the temporal cortex of epileptic patients. We then recorded excitatory and inhibitory postsynaptic currents from layer 5 pyramidal neurons in the cortex of temporal lobe epilepsy patients, before and during selective modulation of $\alpha 4\beta 2^*$ nAChRs by desformylflustrabromine (a selective $\alpha 4\beta 2^*$ positive allosteric modulator). We observed a decrease in both frequency and amplitude of spontaneous excitatory postsynaptic currents, along with an increase in spontaneous inhibitory postsynaptic current frequency. Both these effects were blocked by dihydro-$\beta$-erythroidine, a selective $\alpha 4^*$ antagonist. $\alpha 4\beta 2^*$ activation enhanced the excitability of interneurons (but not of layer 5 pyramidal neurons) by lowering the action potential threshold. Moreover, upon block of action potential propagation by TTX, $\alpha 4\beta 2^*$ activation did not alter miniature inhibitory postsynaptic currents recorded from pyramidal neurons, at the same time as reducing the release at glutamatergic synapses by a $GABA_B$-dependent process.

(Received 23 September 2023; accepted after revision 9 February 2025; first published online 1 March 2025)

**Corresponding author** K. Martinello: Department of Human, Social & Health Sciences, University of Cassino and Southern Lazio, Loc. Folcara, 03043 Cassino, Italy. Email: katiuscia.martinello@unicas.it

## Key points

- Heteromeric nicotinic acetylcholine receptors containing the $\alpha 4$ and $\beta 2$ subunits ($\alpha 4\beta 2^*$ nAChRs) increase GABA release in several regions of the brain.
- We observe an increase of $\alpha 4\beta 2^*$ nAChRs expression in the temporal cortex of patients with temporal lobe epilepsy (TLE, the most represented human focal epilepsy).
- When selectively activated with the positive allosteric modulator desformylflustrabromine (dFBr), $\alpha 4\beta 2^*$ nAChRs increase the frequency of GABA release and decrease the glutamate release onto pyramidal neurons in the layer 5 of human TLE cortex.
- The increase of GABA release is related to an $\alpha 4\beta 2^*$-mediated enhanced excitability of cortical interneurons; instead, the decrease of glutamate release involves a presynaptic $GABA_B$-mediated mechanism, being abolished by a selective $GABA_B$ blocker.
- Our findings show that the activation of $\alpha 4\beta 2^*$ nAChRs induce an increase of the inhibitory tone in human TLE cortex and candidate nicotinic positive allosteric modulators as new pharmacological tools to treat TLE.

## Introduction

Neuronal nicotinic acetylcholine receptors (nAChRs) are expressed at presynaptic and postsynaptic membranes of different kind of synapses throughout the nervous system (Mansvelder et al., 2009; McGehee & Role, 1995). They are pentameric ACh-gated cationic channels differently permeable to $Ca^2$ (Fucile, 2004, 2017; Zoli et al., 2015), mediating fast synaptic transmission and/or regulation of neurotransmitter release (McGehee et al., 1995). The $\alpha$ subunits can assemble into functional homomeric (e.g. $\alpha 7$) or heteromeric nAChRs, assembling with $\beta$ subunits (e.g. $\alpha 4\beta 2^*$, the major heteropentameric nAChR in the brain). nAChRs are activated by ACh released by basal forebrain cholinergic nuclei (BF) (Mesulam et al., 1992), midbrain nuclei (Mesulam et al., 1983) and cholinergic interneurons. Both synaptic and volume cholinergic transmission exist in the neocortex (Disney & Higley, 2020). In prefrontal cortex layer (L)1, 2/3 and 6, optogenetic activation of BF cholinergic fibres triggered synaptic responses mediated by $\alpha 4\beta 2^*$ and/or $\alpha 7$ nAChRs (Hay et al., 2016; Obermayer et al., 2017; Verhoog et al., 2016). When fibres fire at low rates, nicotinic (but not muscarinic) receptors are recruited because they are predominantly located in synapses. At higher fire rates, BF cholinergic neurons also recruit extrasynaptic $\alpha 4\beta 2^*$ nAChRs and muscarinic receptors by spillover (Hay et al., 2016; Kimura et al., 2014). Thus,

the cholinergic system controls postsynaptic response and modulates several time-dependent processes (Arroyo et al., 2014; Obermayer et al., 2017). In human and rodent neocortex, α4β2* nAChRs are responsible for controlling GABA and glutamate release, modulating the excitation/inhibition balance of neocortical circuits (Alkondon et al., 2000). α4β2* nAChRs are associated with the modulation of GABA release from various interneurons in the somatosensory and pre-frontal cortex in rodents (Obermayer et al., 2017). nAChRs contribute to important brain functions, including attention, learning and memory, and mediate the rewarding and aversive effects of nicotine, the major addictive component in tobacco products (Grieder et al., 2019; Hasselmo & Giocomo, 2006; Mansvelder et al., 2009; Sarter et al., 2009; Sciaccaluga et al., 2015). Abnormalities in neuronal nAChRs have been linked to pathological conditions, including cognitive deficits in Alzheimer's and Parkinson's diseases (Gotti et al., 1997; Shimohama et al., 1986), schizophrenia (Koola, 2018) and some forms of epilepsies. Mutations in α4β2* nAChRs can cause autosomal dominant nocturnal frontal lobe epilepsy (i.e. ADNFLE) (Bertrand et al., 2002) but the expression and function of nAChRs appear to be unaffected in animal models of temporal lobe epilepsy (TLE) (Zimmerman et al., 2008). α4β2* nAChRs are associated with the modulation of GABA release from various interneurons in the somatosensory and pre-frontal cortex in rodents (Obermayer et al., 2017). In the present study, we use the selective α4β2* allosteric modulator desformylflustrabromine (dFBr) (Kim et al., 2007) to increase α4β2* nAChR activation in the human neocortex obtained by neurosurgical resection of epileptic foci from TLE patients, aiming to highlight the effects of these receptors in human cortical transmission.

## Methods

### Ethical approval

Each patient provided their written informed consent to use part of the bioptic material for experiments. All the procedures conformed to the latest revision of the Declaration of Helsinki, and the Neuromed Ethics Committee approved the selection processes and procedures (Approval Numbers: ODG.12/28.7.22 and ODG 4/28.2.23) (ClinicalTrials.gov Identifier number: NCT05459090; LES-001).

### Patients

Surgical specimens were obtained from the temporal neocortex of 44 drug-resistant TLE patients and 14 patients affected by temporal tumours (Table 1) operated at the Neuromed Neurosurgery Centre (Pozzilli-Isernia, Italy). The border of each tumour was defined intra-operatively based on neuronavigation using magnetic resonance imaging. After clear identification of the tumour border, microsurgical techniques were used to isolate peritumoural brain tissue.

### Tissue homogenates and TritonX-100 extracts

The dissected frozen temporal cortex of TLE patients and patients affected with temporal tumours were prepared as previously described (Gotti et al., 2006). Protein content of the membranes and 2% Triton X-100 extracts was measured using the bicinchoninic acid protein assay (Pierce, Rockford, IL, USA) with bovine serum albumin as standard.

### Radioligand binding assays

**Binding to membrane.** $(\pm)$-[³H]Epibatidine (³H-Epi) (specific activity of 56–60 Ci mmol⁻¹) and ¹²⁵I-αBungarotoxin (¹²⁵I-αBGTX) (specific activity of 200 Ci mmol⁻¹) were purchased from PerkinElmer (Boston, MA, USA) and non-radioactive ligands were from Tocris (Bristol, UK)

Binding to membranes from the temporal cortex of TLE patients and patients affected with temporal tumours was conducted as previously described (Gotti et al., 2006).

**Binding to solubilized receptors.** To ensure that the α7-containing subtypes did not contribute to ³H-Epi binding, solubilized receptors (present in the extract and immunoprecipitation experiments) were first incubated for 3 h with 1 μM cold αBgtx (Tocris), which specifically binds to α7-nAChR (and thus prevents ³H-Epi binding to these sites). Extracts were labelled with 2 nM ³H-Epi at 4°C and, following overnight incubation, receptors were captured using DEAE-Sepharose™ Fast flow (GE Healthcare, Uppsala, Sweden). The bound receptors were eluted with 1 N NaOH and, after addition of the scintillation mixture (filter count; GE Healthcare), counted in a beta counter. Non-specific binding (averaging 5–10% of total binding) was determined in parallel samples containing 100 nM unlabelled Epi.

### Antibody production and characterization

The subunit-specific polyclonal antibodies (Abs) used have been previously described (Gotti et al., 2006). For the immunoprecipitation experiments, the affinity purified Abs (4 mg mL⁻¹ wet resin) were cross-linked to Protein A Sepharose™ CL-4B (GE Healthcare) by means of 20 mM dimethyl pimelidate (Thermo Fisher

**Table 1. List of patients**

| Patient | Sex | Age (years) | Age at onset of epilepsy (years) | MRI findings | Epileptogenic zone | Surgery | Histopathology |
|---|---|---|---|---|---|---|---|
| 1 | Male | 55 | – | – | – | LES | GB III WHO |
| 2 | Male | 68 | – | – | – | LES | GB IV WHO |
| 3 | Female | 71 | – | L-T-Mes | – | LES | GB IV WHO |
| 4 | Male | 48 | – | R-T-Mes | – | LES | AC II WHO |
| 5 | Female | 49 | – | L-T-Mes | | LES | GB IV WHO |
| 6 | Female | 49 | – | – | – | LES | GB IV WHO |
| 7 | Male | 76 | – | R-T-Mes | – | LES | GB IV WHO |
| 8 | Female | 46 | – | L-T-Mes | – | LES | AS III WHO |
| 9 | Female | 64 | – | R-T-Mes | – | LES | MET |
| 10 | Male | 70 | – | – | – | LES | G I WHO |
| 11 | Male | 55 | – | L-T-Mes | – | LES | AS II WHO |
| 12 | Male | 61 | – | | – | LES | GB IV WHO |
| 13 | Female | 47 | – | R-T-Mes | – | LES | GB IV WHO |
| 14 | Male | 54 | – | – | – | LES | G I WHO |
| 15 | Female | 23 | 1.5 | L-T-Mes | L-T-Mes-Ant | ATL | MES |
| 16 | Female | 23 | 7 | R-T-Mes | R-T-Mes-Ant | ATL | MES |
| 17 | Male | 32 | 9 | L-T-Mes | L-T-Mes | ATL | MES |
| 18 | Female | 19 | 1.5 | R-T-Mes | R-T-Mes | ATL | MES |
| 19 | Female | 36 | 0.5 | R-T-Mes-Ant | R-T-Mes-Ant | ATL | MES |
| 20 | Female | 29 | 6 | L-T-Mes | L-T-Mes | ATL | MES |
| 21 | Female | 39 | 0.5 | R-T-Mes-Ant | R-T-Mes-Ant | ETL | MES |
| 22 | Male | 40 | 3 | L-T-Mes | L-T-Mes | ATL | MES |
| 23 | Female | 36 | 20 | R-T-Mes-Ant | R-T-Mes-Ant | ATL | MES |
| 24 | Female | 29 | 3 | R-T-Mes-Ant | R-T-Mes-Ant | ATL | MES |
| 25 | Female | 49 | 3 | R-T-Mes-Ant | R-T-Mes-Ant | ATL | MES |
| 26 | Male | 45 | 26 | L-T-Mes | L-T-Mes | ATL | MES |
| 27 | Male | 55 | 37 | R-T-Mes-Ant | R-T-Mes-Ant | ATL | MES |
| 28 | Male | 39 | 18 | R-T-Mes-Ant | R-T-Mes-Ant | ATL | MES |
| 29 | Male | 18 | 4 | L-T-Mes | L-T-Mes | ATL | FCD type I b |
| 30 | Male | 25 | 8 | L-T-Mes | L-T-Mes | ATL | MES |
| 31 | Female | 21 | 1 | L-T-Mes | L-T-Mes | ATL | MES |
| 32 | Female | 40 | 7 | R-T-Mes | R-T-Mes | ATL | MES |
| 33 | Female | 27 | 1 | R-T-Mes | R-T-Mes | ATL | MES |
| 34 | Female | 31 | 10 | L-T-Mes | L-T-Mes | ATL | FCD type Ia |
| 35 | Male | 26 | 15 | L-T-Mes | L-T-Mes | ATL | MES |
| 36 | Male | 49 | 3 | R-T-Mes | R-T-Mes | ATL | LES |
| 37 | Male | 31 | 10 | R-T-Mes | R-T-Mes | ATL | FCD type Ia |
| 38 | Female | 36 | 1 | L-T-Mes | L-T-Mes | ATL | DNT |
| 39 | Male | 41 | 2 | L-T-Mes | L-T-Mes | ATL | MES |
| 40 | Male | 51 | 27 | L-T-Mes | L-T-Mes | ATL | MES |
| 41 | Female | 38 | 10 | L-T-Mes | L-T-Mes | ATL | MES |
| 42 | Female | 56 | 6 | L-T-Mes | L-T-Mes | ATL | MES |
| 43 | Male | 55 | 3 | L-T-Mes | L-T-Mes | ATL | CAVERNOUS ANGIOMA |
| 44 | Female | 30 | 6 | R-T-Mes | R-T-Mes | ATL | MES + GB |
| 45 | Female | 60 | 47 | R-T-Mes | R-T-Mes | ATL | MES |
| 46 | Female | 25 | 14 | R-T-Mes | R-T-Mes | ATL | MES |
| 47 | Male | 42 | 7 | R-T-Mes | R-T-Mes | ATL | MES |
| 48 | Female | 42 | 1 | R-T-Mes | R-T-Mes | ATL | MES |
| 49 | Female | 51 | 4 | L-T-Mes | L-T-Mes | ATL | MES |
| 50 | Male | 34 | 13 | R-T-Mes | R-T-Mes | ATL | MES |
| 51 | Female | 53 | 27 | L-T-Mes | L-T-Mes | ATL | MES |
| 52 | Female | 32 | 6 | R-T-Mes | R-T-Mes | ATL | LEAT |

(*Continued*)

**Table 1. (Continued)**

| Patient | Sex | Age (years) | Age at onset of epilepsy (years) | MRI findings | Epileptogenic zone | Surgery | Histopathology |
|---|---|---|---|---|---|---|---|
| 53 | Male | 46 | 20 | R-T-Mes | R-T-Mes | ATL | MES |
| 54 | Male | 25 | 9 | R-T-Mes | R-T-Mes | ATL | DNT |
| 55 | Female | 28 | 15 | L-T-Mes | L-T-Mes | ATL | MES |
| 56 | Male | 32 | 20 | L-T-Mes | L-T-Mes | ATL | MES |
| 57 | Female | 31 | 10 | L-T-Mes | L-T-Mes | ATL | MES |
| 58 | Female | 34 | 7 | L-T-Mes | L-T-Mes | ATL | MES |

Abbreviations: Ant, anterior; AS, astrocytoma; ATL, anterior temporal lobectomy; DNET, Dysembryoplastic neuroepithelial our; ETL, extensive temporal lobectomy; F, frontal; FCD, focal cortical dysplasia; G, ganglioglioma; GB, glioblastoma multiforme; GC, gangliocytoma I° grade WHO; L, left; Lat, lateral; LEAT, long-lasting epilepsy associated with tumour; LES, lesionectomy; MES, hippocampal mesial sclerosis; Mes, mesial; OA, oligoastrocytoma; R, right; T, temporal.

Scientific, Waltham, MA, USA) in accordance with the manufacturer's instructions.

### Immunoprecipitation

Extracts (100 μL), were incubated with 1 μM $\alpha$-bungarotoxin followed by incubation with 2 nM $^3$H-Epi, then incubated overnight with a saturating concentration (10 μg) of affinity-purified anti-subunit IgG (anti-$\alpha 2$,-$\alpha 3$,-$\alpha 4$,-$\alpha 5$,-$\alpha 6$,-$\beta 2$,-$\beta 3$,-$\beta 4$) or control IgG bound to Protein A Sepharose. Immunoprecipitates were recovered by centrifugation (2400 $\boldsymbol{g}$ for 5 min). The level of Ab immunoprecipitation was expressed as fmol of immunoprecipitated receptors mg$^{-1}$ protein.

### Whole-cell recordings from cortical slices

Only temporal cortical tissue from TLE patients was used for the electrophysiological experiments because the corresponding brain tissue from oncological patients was surgically removed in smaller fragments not saving the general synaptic network. Slices (350 μm) were cut tangentially to the outer surface of the cortical specimen, in glycerol-based artificial cerebrospinal fluid (ACSF), with a vibratome (VT 1000S; Leica, Wetzlar, Germany). Slices were placed in a slice incubation chamber at room temperature with oxygenated ACSF and transferred to a recording submerged chamber within 1–12 h after slice preparation. Spontaneous excitatory postsynaptic currents (sEPSCs) and spontaneous inhibitory post-synaptic currents (sIPSCs) were recorded from temporal L5 pyramidal neurons at 24–25°C, using a Multiclamp 700B amplifier (Axon Instruments, Foster City, CA, USA) with a 10 kHz sampling rate and filtered at 2 kHz. Drugs were administered to the cells by perfusion in the bath solution for 5 min. Miniature excitatory postsynaptic currents (mEPSCs) and miniature inhibitory postsynaptic

currents (mIPSCs) were recorded from the same cells, in the presence of TTX (1 μM). APs were recorded in pyramidal cells or interneurons at 24–25°C in the presence of 20 μM CNQX, 50 μM D-AP5, 10 μM bicuculline and 1 μM CGP55485 (20 kHz sampling rate, 2 kHz low pass filter).

### Chemicals and solutions

ACSF composition was (in mM): 125 NaCl, 2.5 KCl, 2 CaCl$_2$, 1.25 NaH$_2$PO$_4$, 1 MgCl$_2$, 26 NaHCO$_3$ and 10 glucose (pH 7.35 with 95% O$_2$/5% CO$_2$). Glycerol-based ACSF solution contained (in mM): 250 glycerol, 2.5 KCl, 2.4 CaCl$_2$, 1.2 MgCl$_2$, 1.2 NaH$_2$PO$_4$, 26 NaHCO$_3$ and 11 glucose (pH 7.35 with 95% O$_2$/5% CO$_2$). For sIPSC recordings, patch pipettes were filled with an intracellular solution containing (in mM): 140 KGluconate, 10 HEPES, 0.5 EGTA and 2 Mg-ATP (pH 7.35 with KOH); holding potential 0 mV. For sEPSC recordings, KGluconate internal solution was replaced by a KCl-based solution, and bicuculline (20 μM) was added to the external solution (−70 mV holding potential). For AP recordings internal solution contained (in mM): 120 KMeSO$_4$, 15 KCl, 10 HEPES, 2 MgCl$_2$, 0.2 EGTA, 2 MgATP and 0.3 mM Tris-GTP. All drugs were purchased from Sigma-Aldrich-Merk (Darmstadt, Germany), Tocris Bioscience (Abingdon UK), Abcam (Cambridge, UK) or Hello Bio (Princeton, NJ, USA) and were freshly prepared before the experiments. Pyramidal neurons and interneurons were identified based on specific criteria: localization (pyramidal L5; interneurons L4), morphology, after-hyperpolarization amplitude (>10 mV for interneurons) (Szegedi et al., 2024) and firing rate.

### Statistical analysis

Data throughout the text represent the mean ± SD. The analysis of spontaneous and miniature PSCs was

performed with Clampfit 10 software (Axon Instruments). The rise time was estimated as the time needed for a 10–90% increase in the peak current response and the decay time as the time needed for a 90–10% decrease in peak current. The mean inhibitory charge of a single synaptic event (*Q*) was measured as the time integral of the synaptic currents. The threshold of single APs elicited by 30 ms current steps from $-70$ mV was measured by differentiating the spike voltage with respect to time (d*V*/d*t*). The resulting values were plotted against the voltage to create a phase plane plot. The threshold was defined as the voltage at the point of deflection for d*V*/d*t* to be greater than zero. To calculate the input resistance (IR), the change of membrane potential induced by a 100 pA depolarizing step was divided by the injected current. APs elicited by 400 ms depolarizing steps were counted at each depolarizing step and reported (Martinello et al., 2015).

As needed, statistical comparisons between groups were made with paired, unpaired *t* tests or one-way ANOVA repeated measures. The power of all performed tests was >0.8 ($\alpha = 0.05$). $P < 0.05$ was considered statistically significant. The effects of drug application on sIPSC and sEPSC kinetics and frequency are summarized in Table 2. The statistical significance of cumulative distributions of amplitudes and inter-event intervals was assessed using the Kolmogorov–Smirnov test with Clampfit 10 software.

## Results

### The expression of heteromeric nAChR subunits is significantly enhanced in the temporal cortex of TLE patients

The expression levels of nAChR subunits in humans and in particular in human epileptic tissues are still not well known. Therefore, we investigated whether the epileptogenic process could lead to a change in nicotinic subunits expression in the human cortex of TLE patients. We measured binding using $^{125}$I $\alpha$BGTX, (selective for $\alpha$7 nAChRs) and $^3$H-Epi (selective for heteromeric nAChRs in the presence of an excess of unlabelled $\alpha$BGTX) in temporal cortices obtained from 14 patients with deep temporal high-grade tumour and 14 with TLE (Fig. 1*A* and *B*). No change in $^{125}$I $\alpha$BGTX binding was detected (60 $\pm$ 26 fmol mg$^{-1}$ protein in peritumoural tissues and 64 $\pm$ 7 fmol mg$^{-1}$ in TLE; $P = 0.712$, unpaired *t* test) (Fig. 1*A*), whereas we found an increase of $^3$H-Epi binding in TLE compared to peritumoural tissue (22 $\pm$ 8 fmol mg$^{-1}$ protein and 14 $\pm$ 8 fmol mg$^{-1}$ protein, respectively, $P = 0.016$, unpaired *t* test) (Fig. 1*B*). We also quantitatively analyzed by immunoprecipitation several heteromeric nAChR subunits in the same tissues. $\alpha$4 and $\beta$2 subunits were the most expressed in the human temporal cortex and significantly enhanced in TLE tissues compared to the peritumoural ones (for $\alpha$4, 17 $\pm$ 6 fmol mg$^{-1}$ protein and 12 $\pm$ 7 fmol mg$^{-1}$ protein, $P = 0.026$, unpaired *t* test; for $\beta$2, 19 $\pm$ 7 fmol mg$^{-1}$ protein and 13 $\pm$ 8 fmol mg$^{-1}$ protein, $P = 0.031$, unpaired *t* test) (Fig. 1*C*).

### The $\alpha$4$\beta$2* nAChR activation decreases sEPSCs recorded from L5 pyramidal neurons

To evaluate the effect of heteromeric nAChRs on the glutamatergic neurotransmission in the human neocortex, we recorded sEPSCs from L5 pyramidal neurons, upon administration of ACh (5 μM for 5 min) (Fig. 2*A–D*). At this concentration, in the presence of the muscarinic antagonist atropine (1 μM), ACh activates mainly the high-affinity $\alpha$4$\beta$2* nAChRs (Palma et al., 2003). To

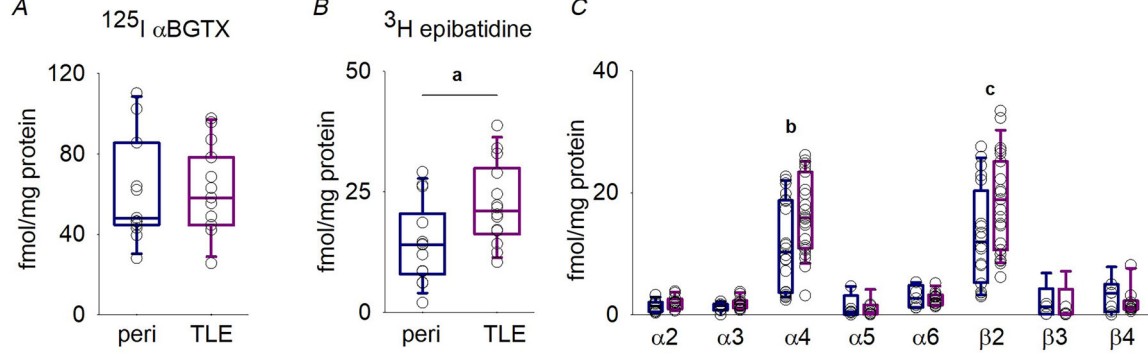

**Figure 1. $\alpha$4 and $\beta$2 subunit expression is enhanced in temporal cortices of TLE patients**
*A*, box plot showing the $^{125}$I $\alpha$BGTX, binding to $\alpha$7-containing receptors in temporal cortex in patients affected with temporal deep tumours (*n* = 12; patients 3–14) or TLE (*n* = 12; patients 17–28). *B*, box plot representing the change in $^3$H-Epi- binding to heteromeric nicotinic receptors in tissues from 14 patients affected with temporal tumour (patients 1–14) and 14 with TLE (patients 15–28; a, *P* = 0.017, unpaired *t* test). *C*, box plot showing the change in the expression of different heteromeric nicotinic receptor (b, *P* = 0.026, unpaired *t* test; c, *P* = 0.031, unpaired *t* test) labelled with $^3$H-Epi and immunoprecipitated by subunit specific antibodies.

**Table 2. Effects of applied drugs on the inhibitory and excitatory post synaptic currents**

| Current | Frequency (Hz) | Amplitude (pA) | Rise time$_{10-90}$ (ms) | Decay time$_{90-10}$ (ms) | Q (pC) |
|---|---|---|---|---|---|
| **sEPSCs** | | | | | |
| Atropine | From 0.8 ± 0.7 | 32 ± 18 | 1.9 ± 0.9 | 4.5 ± 1.4 | 0.13± 0.08 |
| (n = 6) | to 0.9 ± 0.8 | 28 ± 18 | 1.7 ± 0.7 | 4.5 ± 1.3 | 0.11± 0.08 |
| | (P = 0.603, paired t test) | (P = 0.077, paired t test) | (P = 0.602, paired t test) | (P = 0.527, paired t test) | (P = 0.178, paired t test) |
| +ACh | From 1.0 ± 0.6 | 35 ± 23 | 2 ± 1 | 5 ± 3 | 0.23 ± 0.3 |
| (n = 6) | to 1.0 ± 0.5 | 31 ± 21 | 2 ± 1 | 5 ± 3 | 0.20 ± 0.2 |
| | (P = 0.730 paired t test) | (P = 0.075 paired t test) | (P = 0.548 paired t test) | (P = 0.558, paired t test | (P = 0.197, paired t test) |
| +ACh + dFBr | From 1.3 ± 0.7 | 33 ± 18 | 3 ± 2 | 5.2 ± 1.4 | 0.2 ± 0.1 |
| (n = 11) | to 0.4 ± 0.5* | 27 ± 17* | 2.7 ± 1.5 | 5.0 ± 1.6 | 0.2± 0.1 |
| | (*P = 0.002, paired t test) | (*P = 0.034, paired t test) | (P = 0.240, paired t test) | (P = 0.765, paired t test) | (P = 0.114, paired t test) |
| +ACh +dFBr +DHβE | From 0.9 ± 0.5 | 28 ± 7 | 3.5 ± 1.3 | 7 ± 2 | 0.3 ± 0.1 |
| (n = 5) | to 0.9 ± 0.7 | 29 ± 7 | 3.3 ± 1.2 | 7 ± 1 | 0.3 ± 0.1 |
| | (P = 0.499, paired t test) | (P = 0.522, paired t test) | (P = 0.371, paired t test) | (P = 0.182, paired t test) | (P = 0.668, paired t test) |
| **mEPSCs** | | | | | |
| +ACh + dFBr | From 2 ± 1 | 21 ± 1 | 3.5 ± 0.9 | 5.7 ± 0.9 | 0.14 ± 0.02 |
| (n = 4) | to 0.4 ± 0.3 | 21 ± 3 | 3.4 ± 0.7 | 5.7 ± 0.8 | 0.14 ± 0.05 |
| | (*P = 0.048, paired t test) | (P = 0.995, paired t test) | (P = 0.710, paired t test) | (P = 0.837, paired t test) | (P = 0.741, paired t test) |
| +ACh + dFBr +CGP55845 | From 1.6 ±1.6 | 26 ± 7 | 2.5 ± 0.7 | 6.0 ± 0.3 | 0.17 ± 0.06 |
| (n = 6) | to 0.8± 0.7 | 26 ± 5 | 2.7 ±0.8 | 6.7 ± 0.3 | 0.17 ± 0.04 |
| | (P = 0.201, paired t test) | (P = 0.889, paired t test) | (P = 0.422, paired t test) | (P = 0.104, paired t test) | (P = 0.868, paired t test) |
| **sIPSCs** | | | | | |
| Atropine | From 0.9 ± 1.0 | 53 ± 34 | 3.0 ± 1.2 | 14 ± 6 | 0.7 ± 0.4 |
| (n = 6) | to 1 ± 1 | 51 ± 37 | 3.6 ± 2.1 | 13 ± 5 | 0.7 ± 0.4 |
| | (P = 0.528, paired t test) | (P = 0.451, paired t test) | (P = 0.185, paired t test) | (P = 0.185, paired t test) | (P = 0.728, paired t test) |
| +ACh | From 1 ± 1 | 32 ± 14 | 3 ± 1 | 12 ± 6 | 0.4 ± 0.3 |
| (n = 8) | to 2 ± 2 * | 31 ± 12 | 4 ± 1 | 13 ± 6 | 0.3 ± 0.2 |
| | (*P = 0.008, paired t test) | (P = 0.289, paired t test) | (P = 0.165, paired t test) | (P = 0.404, paired t test) | (P = 0.191, paired t test) |
| +ACh +dFBr | From 0.7 ± 1.6 | 44 ± 35 | 6 ± 3 | 18 ± 8 | 0.8 ± 0.5 |
| (n = 8) | to 1.1 ± 1.5* | 41 ± 36 | 6 ± 3 | 17 ± 7 | 0.8 ± 0.5 |
| | (*P = 0.037, paired t test) | (P = 0.169, paired t test) | (P = 0.605, paired t test) | (P = 0.451, paired t test) | (P = 0.095, paired t test) |
| +ACh + dFBr +DHβE | From 0.9 ± 0.2 | 42 ± 8 | 4 ± 1 | 14 ± 4 | 0.6 ± 0.2 |
| (n = 7) | to 1.0 ± 0.3 | 41 ± 10 | 4.2 ± 1.2 | 13 ± 5 | 0.6 ± 0.2 |
| | (P = 0.236, paired t test) | (P = 0.946, paired t test) | (P = 0.631, paired t test) | (P = 0.339, paired t test) | (P = 0.764, paired t test) |
| **mIPSCs** | | | | | |
| +ACh | From 0.6 ± 0.8 | 43 ± 18 | 1.6 ± 0.2 | 13 ± 2 | 0.4 ± 0.1 |
| (n = 3) | to 0.4 ± 0.5 | 37 ± 8 | 2.1 ± 0.4 | 14 ± 1 | 0.5 ± 0.2 |
| | (P = 0.388, paired t test) | (P = 0.404, paired t test) | (P = 0.142, paired t test) | (P = 0.848, paired t test) | (P = 0.399, paired t test) |
| +ACh+ dFBr | From 0.5 ± 0.2 | 46 ± 7 | 4 ± 2 | 15 ± 6 | 0.8 ± 0.5 |
| (n = 5) | to 0.4 ± 0.2 | 39 ± 10 | 4 ± 1 | 12 ± 4 | 0.6 ± 0.3 |
| | (P = 0.633, paired t test) | (P = 0.264, paired t test) | (P = 0.442, paired t test) | (P = 0.147, paired t test) | (P = 0.302, paired t test) |

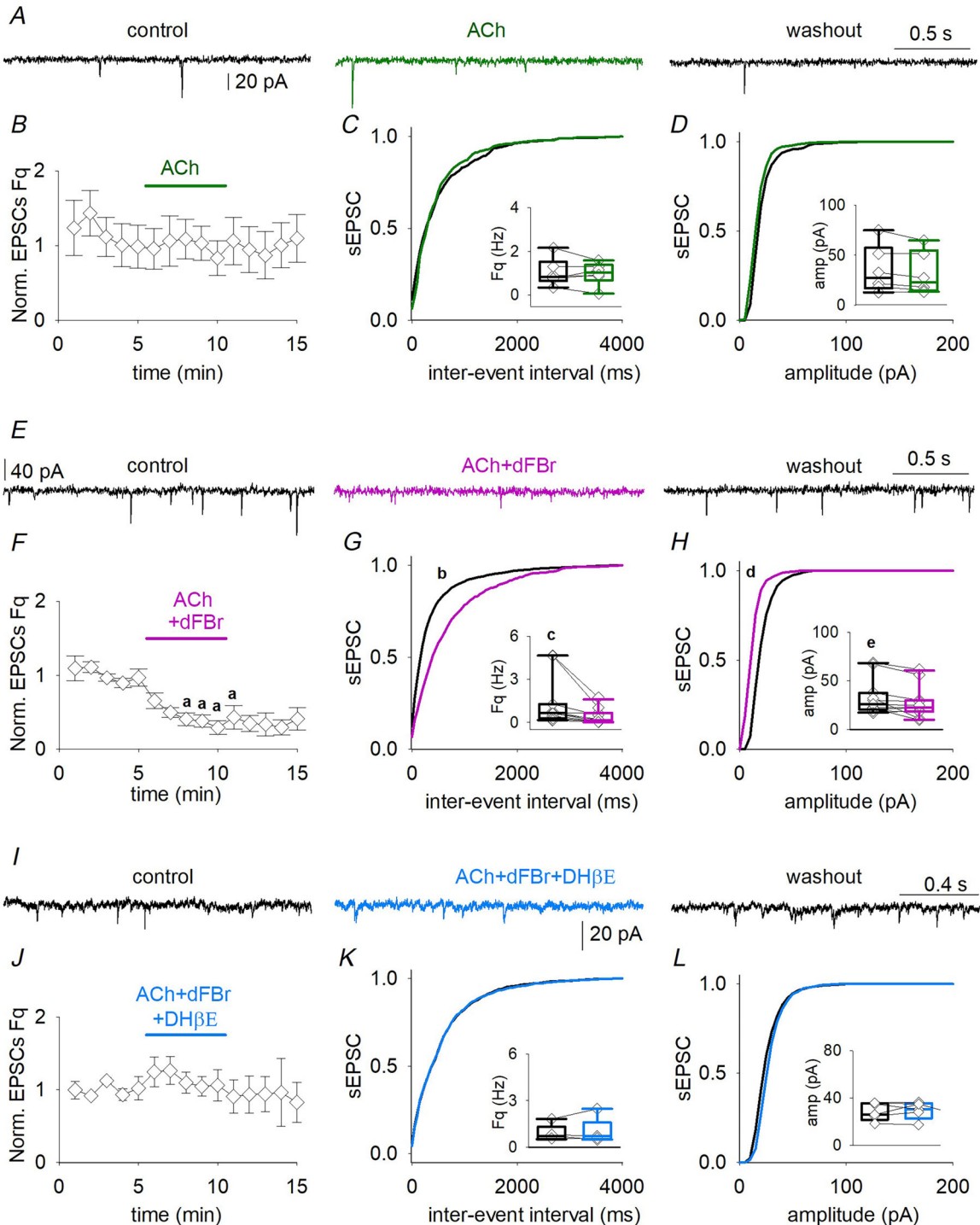

**Figure 2. Activation of heteromeric nicotinic nAChRs reduces glutamate release onto L5 neurons**

*A*, sEPSCs traces recorded from the same neuron (from left to right) in control, when ACh is applied in the presence of atropine (green line) and washout. *B*, time-course of normalized mean sEPSC frequency recorded from six cells (patients 29–31). *C* and *D*, cumulative distribution of sEPSCs inter-event interval (*Ks* = 0.100, *P* = 0.983, Kolmogorov–Smirnov test) and amplitude (*Ks* = 0.199, *P* = 0.781, Kolmogorov–Smirnov test) obtained from same cells as in (*B*) in control (black line) and during ACh and atropine application (green line), inserted box plots showing the mean values and raw data of sEPSCs frequencies and amplitudes (same cells and same contitions). *E*, sEPSCs recorded from the same pyramidal neuron in control condition, when dFBr is coapplied with ACh to slice (pink line) and washout. *F*, time course of normalized sEPSC frequency obtained from 11 neurons during the experiment (patients 29 and 32–34; a, *P* < 0.001, one-way ANOVA repeated measures). *G* and *H*, cumulative distribution of sEPSCs inter-event interval and amplitude obtained from same cells as in (*F*) in control (black line) and during

dFBr, atropine and ACh application (pink line); inserted box plots show the mean values and raw data of sEPSCs frequencies and amplitudes (same cells and same conditions; b, $P < 0.0001$, $Ks = 0.969$, Kolmogorov–Smirnov test; c, $P = 0.002$, paired $t$ test; d, $P = P < 0.0001$ and $Ks = 0.739$, Kolmogorov–Smirnov test; e, $P = 0.002$, paired $t$ test). $I$, sEPSCs recorded from the same neurons in control, when ACh, dFBr and are coapplied with of DHβE (light blue line) and washout. $J$, time course of the normalized sEPSC frequency obtained from five cells (patients 29 and 35). $K$ and $L$, cumulative distribution of sEPSCs inter-event interval ($Ks = 0.313$, $P = 0.061$, Kolmogorov–Smirnov test) and amplitude ($Ks = 0.136$, $P = 0.899$, Kolmogorov–Smirnov test) obtained from same cells and conditions as in $J$, inserted box plots show the mean values and raw data of sEPSCs frequencies and amplitudes. Note, raw data (diamonds) are obtained by averaging the last 3 min values in control and the last three values recorded during drugs application or washout.

exclude possible muscarinic effects, we first assessed that administration of atropine (1 μM for 5 min) had no effect on global sEPSC frequency or kinetics recorded from L5 neurons ($n = 6$) (Fig. 3*A*–*C* and Table 2).

When ACh was applied to L5 pyramidal neurons, no change in sEPSC frequency, amplitude or kinetics was evident ($n = 6$) (Fig. 2*A*–*D* and Table 2). Instead, when we coapplied ACh with dFBr, the sEPSC frequency fell from $1.3 \pm 0.7$ Hz to $0.4 \pm 0.5$ Hz ($n = 11$; $P = 0.002$, paired $t$ test) (Fig. 2*E*–*G* and Table 2). We also observed a decrease in the sEPSCs amplitude in the same cells (from $33 \pm 18$ pA to $27 \pm 17$ pA, $P = 0.034$, paired $t$ test) (Fig. 2*H*). No other change in sEPSCs kinetics was detected (Table 2). Because dFBr acts as a positive allosteric modulator on heteromeric nAChRs and as a mild negative allosteric modulator on homomeric α7 nAChRs (Kim et al., 2007), we investigated whether the effects observed on excitatory neurotransmission were mediated by α4β2* or α7 using the selective α4* antagonist DHβE (20 μM for 5 min) (Fig. 2*I*). In its presence, ACh and dFBr did not affect sEPSC frequency and amplitude (Fig. 2*J*–*L* and Table 2), indicating that the glutamatergic tone recorded from L5 neurons is negatively modulated by α4β2* activation.

A change in glutamatergic transmission could be a result of variation in neuron excitability or quantal release probability. To determine whether the α4β2* activation could result in alterations of pyramidal neuron excitability, we analyzed evoked APs before and during α4β2* receptor positive-modulation with ACh and dFBr ($n = 8$) (Fig. 4*A*–*F*). No significant change in APs frequency ($P = 0.574$, paired $t$ test), threshold ($P = 0.557$, paired $t$ test) or IR ($P = 0.439$, paired $t$ test) was detected (Fig. 4*A*–*F*). The spontaneous AP firing rate, extremely low in our slice preparation as previously reported (Chameh et al., 2021) and, as expected from the low basal frequency of spontaneous excitatory activity, was not affected by nicotinic stimulation (not shown). We then recorded mEPSCs from pyramidal neurons ($n = 4$) (Fig. 4*G*). Coapplication of ACh and dFBr, in the presence of atropine, reduced the mEPSC frequency ($P = 0.048$, paired $t$ test) (Fig. 4*G*–*I*), leaving other kinetic parameters unchanged (Fig. 4*I* and Table 2). Because nAChR activation promotes GABA release and GABA_B receptors (GABA_BRs) activation reduces

glutamate release (Bonanno et al., 1997; Misgeld et al., 1995; Yamada et al., 1999), we recorded mEPSCs in the presence of CGP55845, a selective GABA_BR antagonist (Fig. 4*J*). In this condition, α4β2* activation was no longer able to cause a reduction in mEPSC frequency (Fig. 4*J*–*L* and Table 2). These data, for the first time, indicate the involvement of α4β2* nAChRs in a GABA_B-mediated negative modulation of excitatory synaptic neurotransmission in the human temporal cortex.

### α4β2* nAChR activation enhances spontaneous GABA release onto L5 pyramidal neurons

α4β2* nAChRs promote GABA release in distinct areas of the brain. Thus, we investigated how nicotinic signalling altered GABAergic transmission in the human temporal cortex, by applying ACh (5 μM, in the presence of atropine 1 μM) (Fig. 5). Atropine ($n = 6$; 1 μM for 5 min) did not affect sIPSC frequency or kinetics recorded from L5 neurons (Fig. 3*D*–*F* and Table 2). In pyramidal neurons, ACh administration increased sIPSC frequency ($n = 8$; from $0.7 \pm 0.4$ Hz to $1.7 \pm 0.9$ Hz, $P = 0.008$, paired $t$ test), which quickly recovered during washout (Fig. 5*A*–*C*). No other parameters were affected by ACh (Fig. 5*D* and Table 2). To enhance α4β2* nAChRs activation, ACh was coapplied with dFBr (10 μM) for 5 min (Fig. 5*E*): we still observed an increase of sIPSC frequency ($n = 8$; from $0.7 \pm 0.5$ Hz to $1.1 \pm 0.6$ Hz, $P = 0.0186$, paired $t$ test) (Fig. 5*F* and *G*). dFBr did not affect any other sIPSCs parameters (Fig. 5*H* and Table 2). To confirm that these effects were a result of the activation of α4β2* nAChRs DHβE was applied to L5 pyramidal neurons; in its presence, ACh and dFBr did not affect sIPSC frequency ($n = 7$) (Fig. 5*I*–*L* and Table 2).

### α4β2* nAChR activation increases interneuron excitability in the human neocortex

To unveil the mechanism by which α4β2* activation stimulates GABAergic transmission reaching human L5 pyramidal neurons, we recorded mIPSCs from these cells (Fig. 6). In the presence of TTX, ACh did not affect any mIPSCs parameter ($n = 3$) (Fig. 6*A*–*C* and Table 2);

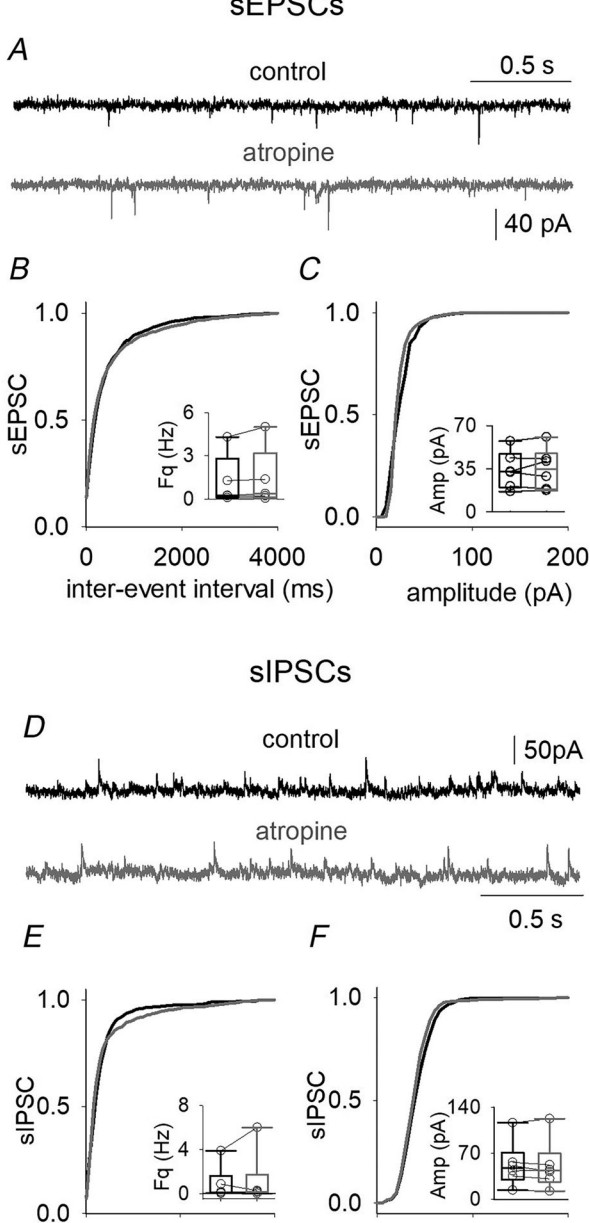

**Figure 3. Atropine application does not affect spontaneous release onto L5 neurons in TLE acute slices**

*A*, examples of sEPSCs recording from a L5 pyramidal neuron in control (black) or during atropine application (grey). *B* and *C*, cumulative distribution of inter-event interval ($Ks = 0.187$, $P = 0.542$, Kolmogorov–Smirnov test) and amplitudes ($Ks = 0.175$, $P = 0.887$, Kolmogorov–Smirnov test) values in control (black line) or during atropine (grey line) recorded in six cells (patients 32, 34 and 29) inserted box plot illustrating mean and raw data obtained from the same cells. *D*, traces representing sIPSCs recording in control and during atropine application (1 μM for 5 min). *E* and *F*, cumulative distribution of inter-event interval ($Ks = 0.304$, $P = 0.078$, Kolmogorov–Smirnov test) and amplitudes ($Ks = 0.394$, $P = 0.106$, Kolmogorov–Smirnov test) values in control (black line) or during atropine (grey line) recorded in six cells (patients 36, 47 and 51).

also, coapplication of ACh with dFBr and atropine was not able to affect mIPSC frequency or amplitude ($n = 5$) (Fig. 6*D–F*). Thus, the $\alpha4\beta2^*$-dependent increase in GABAergic signalling is not caused by a presynaptic mechanism. Another possibility, linked to a wide expression of $\alpha4\beta2^*$ nAChRs in axons, somata and dendrites of interneurons, relies on the ability of these nAChRs to modulate the interneuron excitability (Alkondon et al., 2000). Thus, we recorded APs from L4 interneurons, before and during the application of ACh and dFBr ($n = 4$) (Fig. 6*G*). We observed an increase in AP frequency (at 100 pA steps, from $26 \pm 20$ Hz to $50 \pm 30$ Hz; $P = 0.045$, paired *t* test) (Fig. 6*H*), which was not the result of a change in IR ($P = 0.186$, paired *t* test) (Fig. 6*I*), but instead to a lower AP threshold (from $-50 \pm 2$ mV to $-55 \pm 1$ mV, $P = 0.008$, paired *t* test) (Fig. 6*J–L*).

## Discussion

In the present study, we investigated the expression of heteromeric nAChRs in the human temporal cortex, using brain tissues from patients with TLE or deep brain tumours, along with their functional role in regulating human cortical neurotransmission.

### $\alpha4$ and $\beta2$ subunit expression is enhanced in TLE cortex

To date, a direct correlation between specific allelic variants encoding for $\alpha4$ or $\beta2$ subunits and the ADNFLE is well accepted (Becchetti et al., 2015; Bertrand et al., 2002). Instead, a possible role of $\alpha4\beta2^*$ nAChRs in TLE is not sufficiently investigated: in animal models of TLE, no change in nicotinic expression has been described (Zimmerman et al., 2008) but, to date, limited functional analysis has been performed in TLE tissues. Therefore, we analyzed the composition of nAChRs expressed in 14 temporal cortices obtained from TLE patients and compared them with 14 temporal cortices of patients with deep high-grade non-epileptic tumours (Table 1). In accordance with previous data obtained in the temporal cortex of healthy aged subjects or patients with various forms of dementia (Gotti et al., 2006), $\alpha4$ and $\beta2$ constituted the largely majoritarian populations of heteromeric subunits, whereas the other $\alpha$ subunits comprise between 5% and 15% of the heteromeric nAChRs. By contrast to the homomeric $\alpha7$ subunit, heteromeric subunits are overexpressed in TLE tissue. In particular, the $\alpha4$ and $\beta2$ subunits are increased by ~60% in the human epileptic cortex (Fig. 1). These data, for the first time, highlight that, in TLE patients, during the long epileptogenic process, a reorganization in nAChRs and their subunit composition occurs similarly to GABA and glutamate receptors (Martinello et al., 2018; Mazzuferi

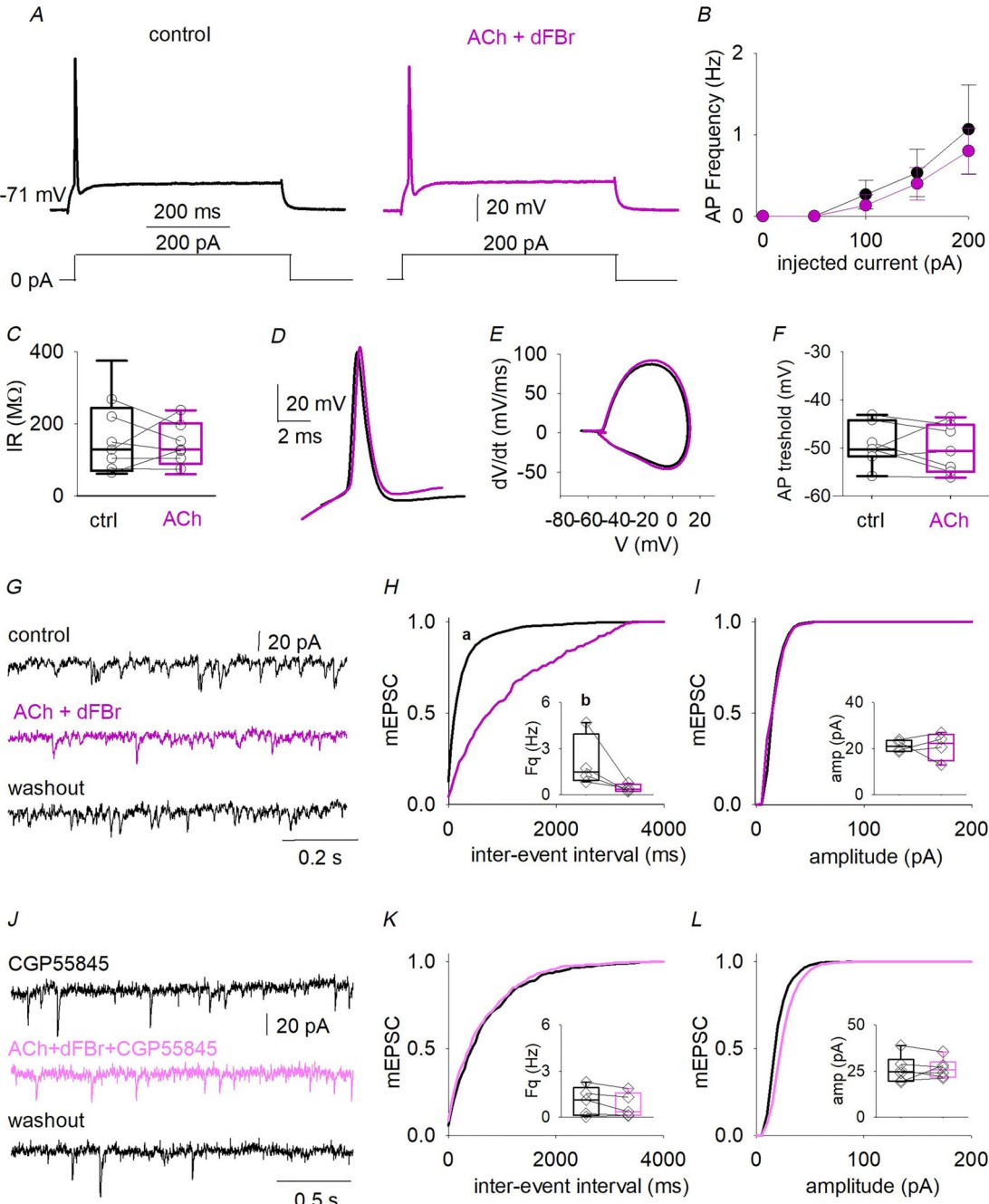

**Figure 4. $\alpha4\beta2$ nAChRs decreased glutamate release probability involving GABA_B receptor activation**
*A*, AP evoked by 200 pA step (400 ms long) in the same pyramidal cell in control (black) and when dFBr and ACh are coapplied (pink). *B*, AP frequency values recorded in seven cells (patients 29, 32 and 35–37). *C*, box plot illustrating input resistance values obtained from the same cells as in (*B*). *D* and *E*, superimposed AP traces recorded from the same neuron in control (black) and during drugs application (pink) and their relative phase-plane plot. *F*, AP threshold values obtained for same cells as in (*B*). *G*, typical records of mEPSCs from the same cell in control (up), when ACh and dFBr are coapplied (middle, pink line) and washout (bottom). *H* and *I*, mIPSCs cumulative distribution of inter-event interval obtained from four pyramidal cells (patients 39–41; a, *P* = 0.0003, *Ks* = 0.789, Kolmogorov–Smirnov test; b, *P* = 0.048, paired *t* test) and amplitudes (*Ks* = 0.216, *P* = 0.971, Kolmogorov–Smirnov test). *J*, mEPSCs recorded from the same neuron in control (up), ACh and dFBr are applied in the presence of CGP55845 (1 μM, light pink, middle, patients 42–46) and during washout (bottom). *K* and *L*, cumulative distribution of inter-event interval *Ks* = 0.371, *P* = 0.120, Kolmogorov–Smirnov test) and amplitudes (*Ks* = 0.197, *P* = 0.878, Kolmogorov–Smirnov test) obtained from six pyramidal cells in control (black line) and in the presence of CGP55845 (light pink line); inserted box plots show the mean values and raw data of mEPSCs frequencies and amplitudes (same cells and same conditions).

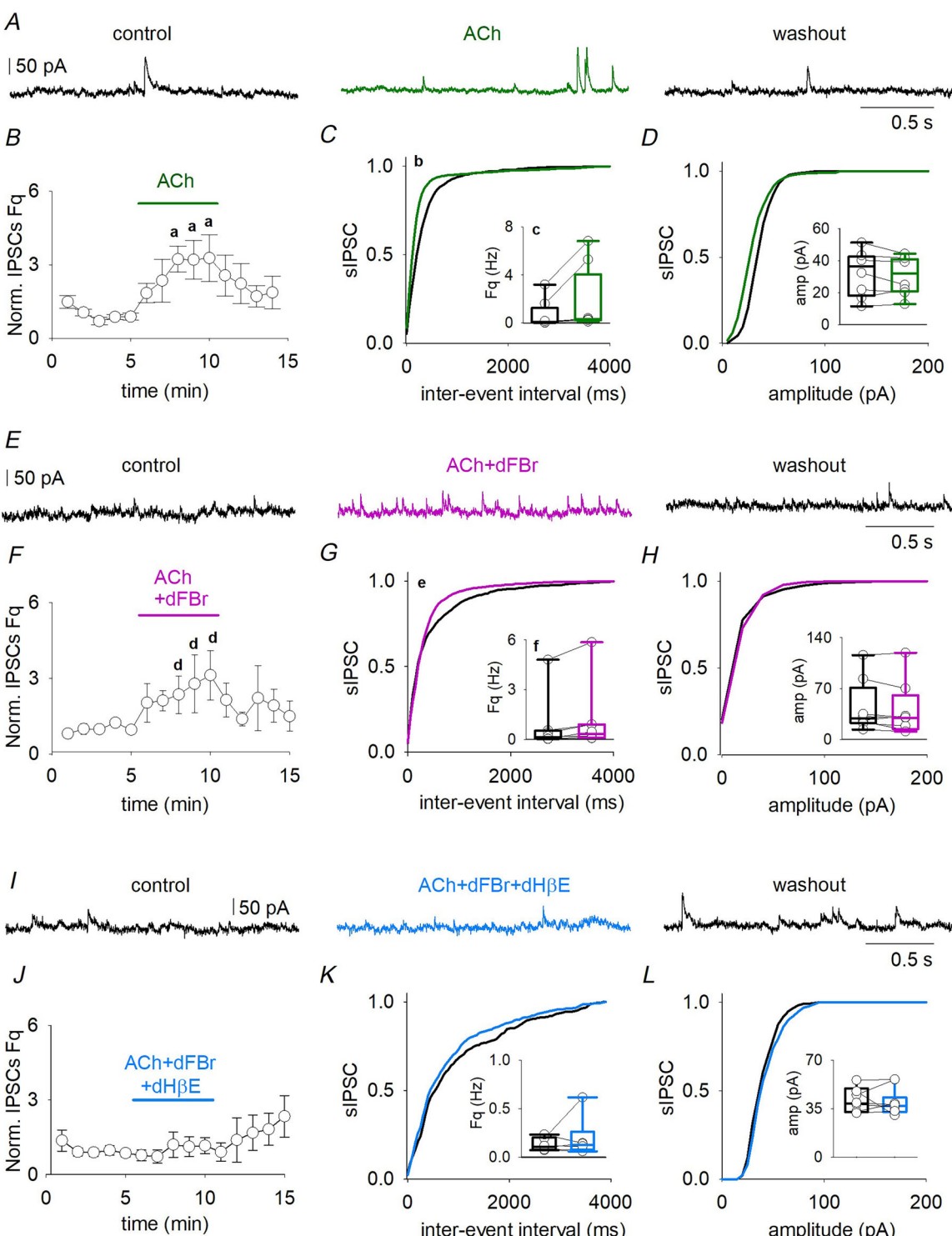

**Figure 5. Activation of α4β2 nAChRs increase GABA release onto L5 neurons**
*A*, typical traces of sIPSCs recorded from L5 neuron (from left to right) in control, when ACh is coapplied with atropine (green line) and during washout. *B*, time-course of normalized mean sIPSC frequency recorded from eight cells (patients 36 and 47–50; a, *P* < 0.001, one way ANOVA repeated measures). *C*, cumulative distribution of inter-event interval of events from the same cells as in (*B*) in control (black line) and during ACh and atropine application (green line), insert box plots showing the mean values of sIPSCs frequencies (same cells and same conditions; b, *P* < 0.001 and *Ks* = 0.954, Kolmogorov–Smirnov test; c, *P* = 0.008). *D*, cumulative distribution of events amplitude from the same cells as in (*B*) (ks = 0.221, *P* = 0.898, Kolmogorov–Smirnov test), insert box plots

showing the mean values of sIPSCs amplitudes (same cells and same conditions). *E*, sIPSCs traces recorded from the same neuron (from left to right) in control, when ACh is coapplied with atropine and dFBr (pink line) and washout. *F*, time course of normalized sIPSC frequency obtained from eight neurons during the experiments (patients 48, 32 and 51–53; d, $P = 0.046$, one-way ANOVA repeated measures). *G*, cumulative distribution of inter-event interval, same cells and in (*F*) and box plots showing the mean values of sIPSCs frequencies in control conditions and during drugs application (e, $P = 0.001$ and ks = 0.873, Kolmogorov–Smirnov test; f, $P = 0.044$, paired *t* test). *H*, cumulative distribution of the amplitudes, same cells as in (*E*) ($Ks = 0.230$, $P = 0.688$, Kolmogorov–Smirnov test). *I*, example traces showing recordings from a pyramidal neuron in control (left), during DHβE plus ACh and atropine bath application (middle, light blue) and during washout (right). *J*, time course of the normalized sIPSC frequency obtained from seven cells (patients 29, 54 and 55) during the experiment. *K* and *L*, cumulative distributions of sIPSCs inter-event interval ($Ks = 0.136$, $P = 0.891$, Kolmogorov–Smirnov test) and amplitudes ($Ks = 0.216$, $P = 0.971$, Kolmogorov–Smirnov test) same cells as in (*J*) and relative inserted box plots showing the mean values of sIPSCs frequencies and amplitudes in control conditions and during drugs application. Note: box plots and circles data are obtained by averaging the last three values in the control condition and the last three values recorded during drug application.

et al., 2010; Miyazaki et al., 2020; Palma et al., 2007; Ren & Curia, 2021; Zilles et al., 1999). The overexpression of α4 and β2 subunits may be a hallmark of TLE, and its functional role in the modulation of cortical activity needs to be analyzed because α4β2* nAChRs can be present at presynaptic locations (Salminen et al., 2004; Zoli et al., 2002) or postsynaptic/somatodendritic locations (Alkondon et al., 2000; Nashmi et al., 2003; Picciotto et al., 2012), modulating both neuronal inputs and outputs.

### Heteromeric nAChR activation reduces glutamatergic transmission by a presynaptic mechanism

α4β2* nAChRs are able to modulate neurotransmitter release at the glutamatergic synapses (Levy et al., 2006; Yang et al., 2020). In L5 pyramidal neurons, we described a DHβE-sensitive reduction in sEPSC frequency and amplitude induced by the activation of heteromeric nAChRs. L5 neurons receive glutamatergic connections from other pyramidal neurons of other cortical layers and present high numbers of recurrent synapses (Kawaguchi, 2017; Yin et al., 2018; Young et al., 2021). Therefore, it is of great relevance to understand how heteromeric nAChRs are involved in the modulation of this cortical excitatory network. We observed that: (i) no α4β2*-mediated currents were detected at L5 somata or proximal dendrites (data not shown); (ii) mEPSC frequency was reduced by heteromeric nAChRs activation in the presence of positive allosteric modulation; and (iii) L5 APs were not affected by the same stimulation. These data indicated a presynaptic mechanism involving α4β2* nAChRs. This finding may appear counterintuitive given that presynaptic nAChRs activation is usually associated with increased neurotransmitter release (Wonnacott, 1997). The apparent paradox was solved by the observation that a GABA$_B$Rs blocker abolishes the nicotinic effect on glutamatergic transmission. Thus, we posit that nAChRs activation locally increases GABA release, which in turn activates GABA$_B$Rs on glutamatergic terminals, producing the observed decrease of glutamate release.

### α4β2* nAChR activation increases GABAergic transmission by a TTX-sensitive mechanism

Indeed, we show here that the activation of heteromeric nAChRs increases GABAergic signalling in the human temporal cortex. According to previous data in the prefrontal cortex (Mansvelder et al., 2009), we did not detect any α4β2*- or α7-mediated current by puffing 5 μm nicotine onto L5 neurons (data not shown): this evidence suggests that α4β2* or α7 nAChRs are not expressed in L5 somata and/or proximal dendrites. Thus, heteromeric nAChRs activation enhanced inhibitory signalling by modulating axonal and/or presynaptic machinery in interneurons and not in postsynaptic L5 membranes. Interestingly, dFBr did not potentiate the ACh-mediated increase of GABA release. This finding is in apparent contrast with the need for dFBr presence to observe a decrease in excitatory synaptic activity. This difference may be the result of distinct functional features of α4β2* nAChRs expressed in diverse interneurons making synaptic connections directly with pyramidal neurons or with excitatory neurons impinging pyramidal neurons. The α4β2*-induced increase of GABA release is probably the result of a direct $Ca^{2+}$ influx through nAChRs (Fucile, 2004, 2017). The $Ca^{2+}$ influx through nAChRs can modulate the neurotransmitter release in several ways: (i) acting presynaptically increasing the $Ca^{2+}$-dependent release probability (Wonnacott, 1997); (ii) acting on $Ca^{2+}$-activated $K^+$ channels (Fuchs & Murrow, 1992); (iii) inducing $Ca^{2+}$ release from intracellular pools (CICR; Mollard et al., 1995); (iv) elevating $[Ca^{2+}]_i$ by depolarizing the cell membrane and opening voltage-dependent $Ca^{2+}$ channels (Zhang & Melvin, 1994); and (v) increasing release via recruitment of voltage-gated $Ca^{2+}$ channels in a TTX-sensitive mechanism (Mansvelder et al., 2009). $Ca^{2+}$ influx in the axonal initial segment can lead to a $Ca^{2+}$-calmodulin modulation of Nav1.6 (Herzog et al., 2003) or to a $Ca^{2+}$-induced Kv7 block (Martinello et al., 2015), in both cases increasing neuronal excitability and firing. In our experiments, mIPSCs were not affected by nAChRs activation, indicating that the nAChR-mediated

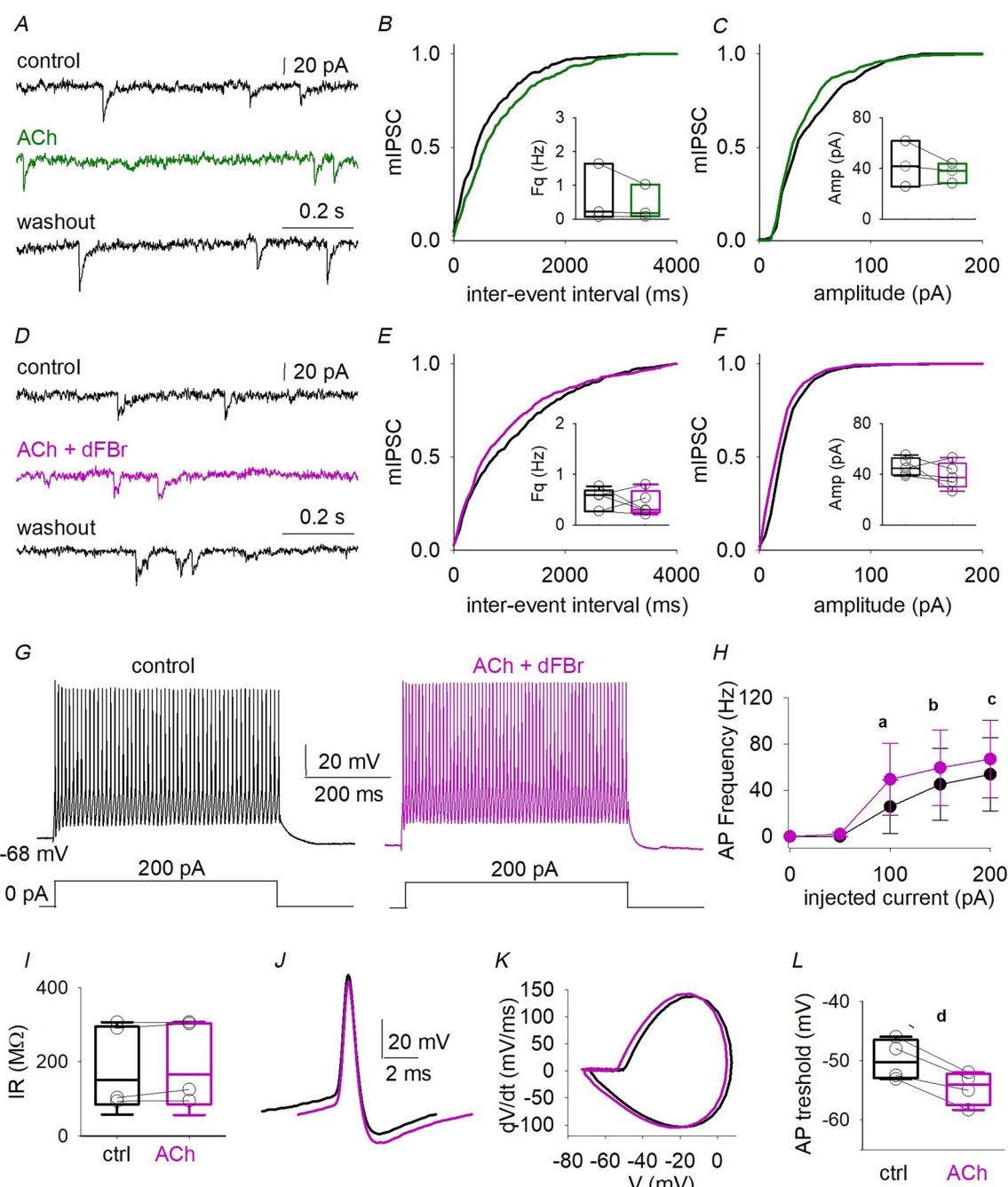

**Figure 6. Activation of $\alpha4\beta2$ nAChRs raises the interneuron firing rate**

*A*, mIPSCs recorded from the same cell in control (up), when atropine and ACh are present in the recording solution (middle, green line) and during the washout (bottom). *B* and *C*, mIPSCs cumulative distribution of inter-event interval ($Ks = 0.175$, $P = 0.290$, Kolmogorov–Smirnov test) and amplitudes ($Ks = 0.243$, $P = 0.361$, Kolmogorov–Smirnov test) obtained from three pyramidal cells (patient 58) in the control condition (black) and during drugs application (green), inserted graphs show the mean values and raw data from same cells. Note, raw data (circle) are obtained by averaging the values recorded in the last 3 min in control, drugs or washout. *D*, mIPSCs recorded from the same cell in control (up), when atropine, ACh and dFBr are coapplied (middle, pink line) and during the washout (bottom). *E* and *F*, mIPSCs cumulative distribution of inter-event interval ($Ks = 0.288$, $P = 0.091$) and amplitudes ($Ks = 0.328$, $P = 0.122$, Kolmogorov–Smirnov test) obtained from five pyramidal cells (patients 37, 56 and 57) in control condition (black) and during drugs application (pink), inserted box plots show the mean values and raw data from same cells. *G*, APs evoked by 200 pA step (400 ms long) in the same fast-spiking interneuron in control (black) and when dFBr, atropine and ACh are coapplied (pink). *H*, mean APs frequencies recorded in four interneurons (patients 36, 32, 29 and 37; a, $P = 0.045$; b, $P = 0.022$; c, $P = 0.046$, paired *t* test), during a current clamp

steps protocol (from 0 to 200 pA, 400 ms long, delta step 50 pA). *I*, values of IR obtained from the same cells as in (*H*). *J* and *K*, typical APs and associated phase plane plots in control (black line) or during drugs (pink line). *L*, APs threshold values obtained from the same cells as in (*H*) and (*I*) (d, *P* = 0.008).

increase of GABA release onto L5 is a TTX-dependent mechanism as previously observed in interneurons (Alkondon et al., 2000). Furthermore, the positive modulation of $\alpha4\beta2^*$ nAChRs enhanced the AP firing rate in L4/5 interneurons by lowering the AP threshold. These data, taken together, indicate that the increase of GABA release onto L5 by $\alpha4\beta2^*$ nAChRs involves an axonal mechanism on GABAergic interneurons. The $Ca^{2+}$ influx in the axon through $\alpha4\beta2^*$ nAChRs may reduce the inactivation state of Nav1.6 by the $Ca^{2+}$-calmodulin pathway, close some $K^+$ channels at the axon initial segment (such as $K_V7$; Martinello et al., 2015) or recruit voltage-gated $Ca^{2+}$ channels (Mansvelder et al., 2009), lowering the APs threshold.

Our data describe, for the first time, a coexisting two-faced modulation of excitatory and inhibitory synaptic transmission in the human epileptic temporal cortex. The overexpression of these nAChRs and their function, modulating GABAergic and glutamatergic transmission in opposite ways, leads to an increase of the cortical inhibitory tone and could be a compensatory mechanism that epileptic tissue hires to prevent and reduce seizures. In particular, the $\alpha4\beta2^*$ nAChRs-induced reduction of glutamate release may partially compensate for the use-dependent run-down of $GABA_A$R-mediated currents (Roseti et al., 2008) and the loss of the functional cross-talk between $GABA_A$Rs and $GABA_B$Rs in L5 pyramidal neurons of the human epileptic cortex (Martinello et al., 2018) On the other hand, positive allosteric modulation of $\alpha4\beta2^*$ nAChRs with molecules such as dFBr could be taken into account to develop new pharmacological strategies to treat drug-resistant TLE. dFBr has already been shown to be well tolerated, in both human and animal models, and to reduce nicotine withdrawal symptoms caused by tobacco smoke (Hamouda et al., 2018; Liu, 2013), to attenuate a compulsive-like behaviour in an animal model (Mitra et al., 2017) and to potentiate nicotine-induced anti-allodynic response in a mouse model of neuropathic pain (Bagdas et al., 2018). All these lines of evidence support the proposal that positive allosteric modulators selective for heteromeric nAChRs may be used for TLE treatment in the future.

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

## Additional information

### Data availability statement

The data that support the findings of this study are available from the corresponding author upon reasonable request.

### Competing interests

None declared.

### Author contributions

K.M performed the experiments and the analyses, wrote the original draft, and reviewed and edited the document table

and figures. A.M, S.C, G.D.G and V.E reviewed and edited the final document. M.Z and C.G performed the experiments and reviewed and edited the final document. S.F conceptualized the study, advised in the execution of the experiments and analyses, wrote the original draft and reviewed and edited the final document with the support of all the others authors. All authors contributed to the article and approved the final version of the manuscript submitted for publication. All authors agree to be accountable for the content of the work. All persons designated as authors qualify for authorship and all those who qualify for authorship are listed.

## Funding

This research was funded by the European Union – Next Generation EU – NRRP M6C2 – Investment 2.1 Enhancement and strengthening of biomedical research in the NHS (PNRR-MAD-2022-12376434).

## Keywords

desformylflustrabromine, excitatory postsynaptic currents, inhibitory postsynaptic currents, L5 pyramidal cells, positive allosteric modulator, temporal lobe epilepsy

## Supporting information

Additional supporting information can be found online in the Supporting Information section at the end of the HTML view of the article. Supporting information files available:

**Peer Review History**

