## [Peer Review History · The Journal of Physiology]

$\alpha 4\beta 2^*$ nicotinic acetylcholine receptors drive human temporal glutamate/GABA balance toward inhibition

Katiuscia Martinello, Addolorata Mascia, Sara Casciato, Giancarlo Di Gennaro, Vincenzo Esposito, Michele Zoli, Cecilia Gotti, and Sergio Fucile

DOI: 10.1113/JP285689

Corresponding author(s): Katiuscia Martinello (katiuscia.martinello@neuromed.it)

Review Timeline:

Submission Date:	23-Sep-2023
Editorial Decision:	22-Mar-2024
Revision Received:	04-Dec-2024
Editorial Decision:	15-Jan-2025
Revision Received:	24-Jan-2025
Accepted:	09-Feb-2025

Senior Editor: Katalin Toth

Reviewing Editor: Katalin Toth

Transaction Report:

Dear Dr Martinello,

Re: JP-RP-2023-285689 " $\alpha 4\beta 2$ * nicotinic acetylcholine receptors drive human temporal glutamate/GABA balance toward inhibition" by Katuscia Martinello, Addolorata Mascia, Sara Casciato, Giancarlo Di Gennaro, Vincenzo Esposito, Michelle Zoli, Cecilia Gotti, and Sergio Fucile

Thank you for submitting your manuscript to The Journal of Physiology. It has been assessed by a Reviewing Editor and by an expert referee and we are pleased to tell you that it is potentially acceptable for publication following satisfactory major revision.

LANGUAGE EDITING AND SUPPORT FOR PUBLICATION: If you would like help with English language editing, or other article preparation support, Wiley Editing Services offers expert help, including English Language Editing, as well as translation, manuscript formatting, and figure formatting at www.wileyauthors.com/eoo/preparation. You can also find resources for Preparing Your Article for general guidance about writing and preparing your manuscript at www.wileyauthors.com/eoo/prepresources.

REVISION CHECKLIST:

Please upload two versions of your manuscript text: one with all relevant changes highlighted and one clean version with no changes tracked. The manuscript file should include all tables and figure legends, but each figure/graph should be uploaded as separate, high-resolution files. The journal is now integrated with Wiley's Image Checking service. For further details, see: <https://www.wiley.com/en-us/network/publishing/research-publishing/trending-stories/upholding-image-integrity-wileys->

image-screening-service

We look forward to receiving your revised submission.

Yours sincerely,

Katalin Toth
Senior Editor
The Journal of Physiology

REQUIRED ITEMS FOR REVISION

- Author photo and profile. First or joint first authors are asked to provide a short biography (no more than 100 words for one author or 150 words in total for joint first authors) and a portrait photograph. These should be uploaded and clearly labelled together in a Word document with the revised version of the manuscript. See Information for Authors for further details.

- You must start the Methods section with a paragraph headed Ethical Approval. If experiments were conducted on humans, confirmation that informed consent was obtained, preferably in writing, that the studies conformed to the standards set by the latest revision of the Declaration of Helsinki and that the procedures were approved by a properly constituted ethics committee, which should be named, must be included in the article file. If the research study was registered (clause 35 of the Declaration of Helsinki), the registration database should be indicated, otherwise the lack of registration should be noted as an exception (e.g. The study conformed to the standards set by the Declaration of Helsinki, except for registration in a database). For further information see: <https://physoc.onlinelibrary.wiley.com/hub/human-experiments>.

- Please upload separate high-quality figure files via the submission form.

- Your paper contains Supporting Information of a type that we no longer publish, including supplementary tables and figures. Any information essential to an understanding of the paper must be included as part of the main manuscript and figures. The only Supporting Information that we publish are video and audio, 3D structures, program codes and large data files. Your revised paper will be returned to you if it does not adhere to our Supporting Information Guidelines.

- Papers must comply with the Statistics Policy: https://jp.msubmit.net/cgi-bin/main.plex?form_type=display_requirements#statistics.

In summary:

- If $n \leq 30$, all data points must be plotted in the figure in a way that reveals their range and distribution. A bar graph with data points overlaid, a box and whisker plot or a violin plot (preferably with data points included) are acceptable formats.

- If $n > 30$, then the entire raw dataset must be made available either as supporting information, or hosted on a not-for-profit repository, e.g. FigShare, with access details provided in the manuscript.

- 'n' clearly defined (e.g. x cells from y slices in z animals) in the Methods. Authors should be mindful of pseudoreplication.

- All relevant 'n' values must be clearly stated in the main text, figures and tables.

- The most appropriate summary statistic (e.g. mean or median and standard deviation) must be used. Standard Error of the Mean (SEM) alone is not permitted.

- Exact p values must be stated. Authors must not use 'greater than' or 'less than'. Exact p values must be stated to three significant figures even when 'no statistical significance' is claimed.

- Please include an Abstract Figure file, as well as the Figure Legend text within the main article file. The Abstract Figure is a piece of artwork designed to give readers an immediate understanding of the research and should summarise the main conclusions. If possible, the image should be easily 'readable' from left to right or top to bottom. It should show the physiological relevance of the manuscript so readers can assess the importance and content of its findings. Abstract Figures should not merely recapitulate other figures in the manuscript. Please try to keep the diagram as simple as possible and without superfluous information that may distract from the main conclusion(s). Abstract Figures must be provided by authors no later than the revised manuscript stage and should be uploaded as a separate file during online submission labelled as File Type 'Abstract Figure'. Please also ensure that you include the figure legend in the main article file. All Abstract Figures should be created using BioRender. Authors should use The Journal's premium BioRender account to export high-resolution images. Details on how to use and access the premium account are included as part of this email.

- Please include a full title page as part of your main article (Word) file, which should contain the following: title, authors, affiliations, corresponding author name and contact details, keywords, and running title.

EDITOR COMMENTS

While the paper does have some good findings, the referee raises several major concerns that would require an extensive revision and re-analysis. In the current form it isn't possible to accept.

Apologies for the delay in a decision.

Senior Editor:

In this manuscript Martinello et al., investigates how nicotinic receptor expression is altered in TLE patients and how this could influence the network. Towards this aim they compare expression levels using binding studies and immunoprecipitation. In the rest of the manuscript, they use TLE resected tissue to investigate excitatory and inhibitory synaptic signals and evaluate the excitability of pyramidal cells and interneurons. The study is addressing an important question, currently we have very little information about cellular level changes in human epileptic samples. While the general premise of the work is quite exciting there are several points that would need to be significantly improved in order to give a clearer picture on how changes in nAChR expression could potentially be involved in the pathogenesis.

1. The sEPSCs and mEPSCs are affected by ACh + dFBr, but not ACh alone. sIPSC frequency is influenced by Ach alone and dFBr does not make much difference. mIPSCs and interneuron excitability is investigated only in ACh + dFBR and not in ACh alone. This experimental design seems to be confusing. Why not compare ACh with ACh+dFBr in all these conditions?
2. Surprisingly, after Fig. 1. the peritumoral tissue is not used as a control. This would be essential to determine whether the effects described in subsequent figures are related to TLE.
3. Why is ACh effective to influence IPSCs, while only ACh+dFBr is shown to effect EPSCs?
4. How was the spontaneous activity (spontaneous firing) of the recorded neurons influenced by ACh/ ACh+dFBr?
5. How were evoked synaptic currents influenced by ACh/ ACh+dFBr?
6. Does ACh/ ACh+dFBr change the threshold for triggering seizure-like activity in TLE and peritumoral tissue?

REFEREE COMMENTS

Referee #1:

The authors have conducted a strong assessment of the modulation of neurotransmitter release by nicotinic receptors in human cortical tissue slices from epileptic surgical resections. Investigation of human synaptic physiology in normal and diseased tissue is rare and the findings here will be of broad interest. The observations that nAChR expression is altered in epileptic tissue is interesting and the shift in synaptic modulation by those receptors could contribute to enhanced excitability. Some concerns with the manuscript are outlined below.

Major concerns:

- 1) One concern that is not addressed is the extent of neuronal death occurring during the time between resection, sectioning and recording, and whether this affects individual neuron types differentially. This could lead to under- or over-representation of the synaptic modulation explored here.
- 2) There are many grammatical errors, awkward wording and unclear sentences throughout the manuscript. Careful editing will be required for publication.
- 3) The aCSF solution contains a bicarbonate buffer that requires physiological CO₂ to maintain neutral pH. There is no mention of saturating the aCSF recording media with 95% O₂ and 5% CO₂. If this was not done, then the pH of the external solution will not be maintained at 7.4
- 4) The statistical tests used to assess differences between treatment groups should be stated along with the p values for each experiment.
- 5) There is no mention of the criteria for identifying 'peritumoral' tissues - was this based solely on histology? Were markers used or gross histological features? What distance from the tumor border was used?
- 6) Criteria for identifying interneurons in the preparation should be explained clearly.
- 7) This may not be particularly relevant to the current study, but it would be interesting to assess the nAChR subunits that are immunoprecipitated in the BGT binding assay

Minor:

Last sentence of the Abstract: 'Inhibitory miniature currents' should read miniature 'inhibitory postsynaptic currents'

DHbE is typically abbreviated with a capital 'D'

Axis labels should use sEPSC or mEPSC rather than EPSC. N. Count is the confusing axis label on all the cumulative distributions - please consider a more interpretable label. Also, I recommend spelling out the axis labels wherever possible.

END OF COMMENTS

Dear Sirs and Madams,

We thank the Senior Editor and the Reviewer for their comments that helped us substantially to improve the Manuscript.

We aim to start this reply with two main considerations:

- 1) we have very low amount of human tissue (in this last 8-months period from first submission we received temporal brain tissue from only two patients undergoing surgery), so we had to give priority to selected experiments.
- 2) Senior Editor asks, rightly, for data concerning the overall influence of nicotinic stimulation on the cortical output (pyramidal firing, seizure-like activity), but this kind of activity is spontaneously very low or absent in our preparation, as detailed below, and we have not a sufficient amount of brain tissue to parallel the present data with new experiments concerning evoked synaptic currents or evoked seizure-like activity. Evoked activity and parallel behavioural experiment in animal models could be the main topic of the next paper.

Senior Editor:

In this manuscript Martinello et al., investigates how nicotinic receptor expression is altered in TLE patients and how this could influence the network. Towards this aim they compare expression levels using binding studies and immunoprecipitation. In the rest of the manuscript, they use TLE resected tissue to investigate excitatory and inhibitory synaptic signals and evaluate the excitability of pyramidal cells and interneurons. The study is addressing an important question, currently we have very little information about cellular level changes in human epileptic samples. While the general premise of the work is quite exciting there are several points that would need to be significantly improved in order to give a clearer picture on how changes in nAChR expression could potentially be involved in the pathogenesis.

1. The sEPSCs and mEPSCs are affected by ACh + dFBr, but not ACh alone. sIPSC frequency is influenced by Ach alone and dFBr does not make much difference. mIPSCs and interneuron excitability is investigated only in ACh + dFBR and not in ACh alone. This experimental design seems to be confusing. Why not compare ACh with ACh+dFBr in all these conditions?

We added new data about the modulation of mIPSCs with ACh alone recorded from pyramidal neurons, as requested (line 274 and Fig. 6A-C). We could not provide data about interneuron excitability with ACh alone.

2. Surprisingly, after Fig. 1. the peritumoral tissue is not used as a control. This would be essential to determine whether the effects described in subsequent figures are related to TLE.

Peritumoral tissue is not adequate as a control for synaptic network. Peritumoral tissue is removed during surgery in small pieces which do not allow to obtain slices, and do not allow to study synaptic activity in a comparable experimental condition as TLE tissue. We added in the Method section a sentence explaining this problem (lines 149-151). We changed a sentence in the

Discussion, stating that our work is aimed at understanding the role of nicotinic receptors in the modulation of cortical activity (line 310), and not their role in the pathology.

3. Why is ACh effective to influence IPSCs, while only ACh+dFBr is shown to effect EPSCs?

We added in Discussion two sentences to highlight the possibility that distinct interneurons may express distinct nAChR with specific sensibility to ACh and dFBr (lines 341-344).

4. How was the spontaneous activity (spontaneous firing) of the recorded neurons influenced by ACh/ ACh+dFBr?

We have done new gap-free experiments on pyramidal neurons, with ACh or ACh+dFBr . In particular, the spontaneous firing was absent or extremely low recording for L5 pyramidal neurons in slices, consistently with low basal glutamatergic signalling and with reports in the literature, both for mouse (70% of L5 pyramidal neurons with no spontaneous activity; Mao et al, 2001) and in humans (near 0 Hz frequency of spontaneous firing rate; Chameh et al., 2021), and, in our hand, was not affected by nicotinic stimulation. We added a sentence in the Result section, illustrating these data (not shown in figures; lines 240-242).

5. How were evoked synaptic currents influenced by ACh/ ACh+dFBr?

We did not study evoked synaptic currents for two main reasons: i) limited availability of human brain tissue; ii) lack of ACh-related kinetic differences in spontaneous synaptic activity suggesting a postsynaptic mechanism.

6. Does ACh/ ACh+dFBr change the threshold for triggering seizure-like activity in TLE and peritumoral tissue?

Spontaneous seizure-like activity is absent in our human temporal cortex preparation. For the same reasons explained in the previous points 4 and 5, we did not study in slices evoked seizure-like activity. The effect on seizure-like activity of promoting nicotinic signalling with dFBr will be better analyzed by behavioural and electrophysiological studies in living animals from an experimental model of TLE, in a future work.

REFEREE COMMENTS

Referee #1:

The authors have conducted a strong assessment of the modulation of neurotransmitter release by nicotinic receptors in human cortical tissue slices from epileptic surgical resections. Investigation of

human synaptic physiology in normal and diseased tissue is rare and the findings here will be of broad interest. The observations that nAChR expression is altered in epileptic tissue is interesting and the shift in synaptic modulation by those receptors could contribute to enhanced excitability. Some concerns with the manuscript are outlined below.

Major concerns:

1) One concern that is not addressed is the extent of neuronal death occurring during the time between resection, sectioning and recording, and whether this affects individual neuron types differentially. This could lead to under- or over-representation of the synaptic modulation explored here.

Brain tissue is already damaged (cell loss) by the enduring pathology before surgery and slice preparation. Selective vulnerability of neurons in defined hippocampal and cortical areas has been described in TLE patients, but to our knowledge, there is no information about the differential vulnerability of specific neuronal subtypes to the slicing procedure in TLE tissue. We started the standard slicing procedure within 15 minutes from surgery, minimizing oxygen deprivation and excitotoxic damage. Furthermore, both glutamatergic and GABAergic synaptic activity is stable in the time window in which we use the slices, indicating a stable survival of neurons after the cut. Indeed, to analyse the effect of cutting on selective neuronal subtype vulnerability it would be relevant to compare cut and uncut tissue after fixation, using immunohistological methods. We had not sufficient amount of human brain tissue to do this characterization, which, to our knowledge, is not present in the literature, both for human and mammalian tissues.

2) There are many grammatical errors, awkward wording and unclear sentences throughout the manuscript. Careful editing will be required for publication.

We carefully checked the English grammar and wording. We edited the text to improve clarity.

3) The aCSF solution contains a bicarbonate buffer that requires physiological CO₂ to maintain neutral pH. There is no mention of saturating the aCSF recording media with 95% O₂ and 5% CO₂. If this was not done, then the pH of the external solution will not be maintained at 7.4

We use routinely 95% O₂ and 5% CO₂. We added this information in the Method section (lines 163 and 165).

4) The statistical tests used to assess differences between treatment groups should be stated along with the p values for each experiment.

We added this information in the figure legends, table, and main text.

5) There is no mention of the criteria for identifying 'peritumoral' tissues - was this based solely on histology? Were markers used or gross histological features? What distance from the tumor border was used?

We obtained routinely histological confirmation of tumoral nature of the lesion, as specified in the table. We added a sentence in the Method section illustrating the procedure to obtain peritumoral tissue (lines 104-106). No biological markers were used in the procedure. The peritumoral tissue was removed based only on clinical criteria, so the distance between the used tissue and the tumour border may vary in different patients.

6) Criteria for identifying interneurons in the preparation should be explained clearly.

We added a sentence in the Method section, to better explain how interneurons were identified (lines 172-175).

7) This may not be particularly relevant to the current study, but it would be interesting to assess the nAChR subunits that are immunoprecipitated in the BGT binding assay

The possible interaction between different subunits could be a very interesting point for a future study. In particular, it would be highly interesting to understand if $\beta 2$ subunits can interact directly with $\alpha 7$ subunit in human brain.

Minor:

Last sentence of the Abstract: 'Inhibitory miniature currents' should read miniature 'inhibitory postsynaptic currents'

We emended the text as requested.

DHbE is typically abbreviated with a capital 'D'

We emended the text as requested.

Axis labels should use sEPSC or mEPSC rather than EPSC. N. Count is the confusing axis label on all the cumulative distributions – please consider a more interpretable label. Also, I recommend spelling out the axis labels wherever possible.

We changed the axis labels as requested.

Dear Dr Martinello,

Re: JP-RP-2024-285689R1 " $\alpha 4\beta 2$ * nicotinic acetylcholine receptors drive human temporal glutamate/GABA balance toward inhibition" by Katuscia Martinello, Addolorata Mascia, Sara Casciato, Giancarlo Di Gennaro, Vincenzo Esposito, Michele Zoli, Cecilia Gotti, and Sergio Fucile

Thank you for submitting your revised Research Article to The Journal of Physiology. It has been assessed by the original Reviewing Editor and Referees and has been well received. Some final revisions have been requested.

REVISION CHECKLIST:

We look forward to receiving your revised submission.

Yours sincerely,

Katalin Toth
Senior Editor
The Journal of Physiology

REQUIRED ITEMS

- Papers must comply with the Statistics Policy: https://jp.msubmit.net/cgi-bin/main.plex?form_type=display_requirements#statistics.

In summary:

- If n {less than or equal to} 30, all data points must be plotted in the figure in a way that reveals their range and distribution. A bar graph with data points overlaid, a box and whisker plot or a violin plot (preferably with data points included) are acceptable formats.
- If $n > 30$, then the entire raw dataset must be made available either as supporting information, or hosted on a not-for-profit repository, e.g. FigShare, with access details provided in the manuscript.
- 'n' clearly defined (e.g. x cells from y slices in z animals) in the Methods. Authors should be mindful of pseudoreplication.
- All relevant 'n' values must be clearly stated in the main text, figures and tables.
- The most appropriate summary statistic (e.g. mean or median and standard deviation) must be used. Standard Error of the Mean (SEM) alone is not permitted.
- Exact p values must be stated. Authors must not use 'greater than' or 'less than'. Exact p values must be stated to three significant figures even when 'no statistical significance' is claimed.

EDITOR COMMENTS

Please attend to the minor revision suggested by the Reviewer.

REFEREE COMMENTS

Referee #1:

The authors have addressed my concerns and the manuscript is improved.

Minor suggestion, please check the methods section line 172 description of identification of cell types, I would expect the AHP in cortical interneurons to be less than 10 mV, but the description says '>10 mV'.

END OF COMMENTS

Dear Sirs,

We thank the Senior Editor and the Reviewer for the good reception of the paper.

Following the suggestion of Reviewer, we have slightly changed the text as detailed below.

Minor suggestion, please check the methods section line 172 description of identification of cell types, I would expect the AHP in cortical interneurons to be less than 10 mV, but the description says '>10 mV'.

To increase clarity, we have changed, in the Method section (now line 174) the term “after-hyperpolarization” with “after-hyperpolarization amplitude” and we have added a new Reference (Szegedi et al. J Biotechnol 20:1-12) in which it is possible to observe that after-hyperpolarization amplitude data recorded in human cortical interneurons mainly range from 10 mV to 20 mV. We think that now the indication >10 mV is more comprehensible.

The only other modification in the text is in the Acknowledgement section: we have added the serial number of the European grant supporting our research activity.

In the Abstract figure and in the Abstract figure legend, we changed “alpha4beta2” into $\alpha 4\beta 2^*$

Thank you again.

Dear Dr Martinello,

Re: JP-RP-2025-285689R2 " $\alpha 4\beta 2$ * nicotinic acetylcholine receptors drive human temporal glutamate/GABA balance toward inhibition" by Katuscia Martinello, Addolorata Mascia, Sara Casciato, Giancarlo Di Gennaro, Vincenzo Esposito, Michele Zoli, Cecilia Gotti, and Sergio Fucile

We are pleased to tell you that your paper has been accepted for publication in The Journal of Physiology.

Yours sincerely,

Katalin Toth
Senior Editor
The Journal of Physiology

If you would like to receive our 'Research Roundup', a monthly newsletter highlighting the cutting-edge research published in The Physiological Society's family of journals (The Journal of Physiology, Experimental Physiology, Physiological Reports, The Journal of Nutritional Physiology and The Journal of Precision Medicine: Health and Disease), please click this link, fill in your name and email address and select 'Research Roundup':
<https://www.physoc.org/journals-and-media/membernews>

- You can help your research get the attention it deserves! Check out Wiley's free Promotion Guide for best-practice recommendations for promoting your work at: www.wileyauthors.com/eoo/guide. You can learn more about Wiley Editing Services which offers professional video, design, and writing services to create shareable video abstracts, infographics, conference posters, lay summaries, and research news stories for your research at: www.wileyauthors.com/eoo/promotion.
